# Noise-tolerant fair classification

**Alexandre Lamy**[*]
Columbia University
a.lamy@columbia.edu

**Ziyuan Zhong**[*]
Columbia University
ziyuan.zhong@columbia.edu

**Aditya Krishna Menon**
Google
adityakmenon@google.com

**Nakul Verma**
Columbia University
verma@cs.columbia.edu

## Abstract

Fairness-aware learning involves designing algorithms that do not discriminate with respect to some sensitive feature (e.g., race or gender). Existing work on the problem operates under the assumption that the sensitive feature available in one's training sample is perfectly reliable. This assumption may be violated in many real-world cases: for example, respondents to a survey may choose to conceal or obfuscate their group identity out of fear of potential discrimination. This poses the question of whether one can still learn fair classifiers given *noisy* sensitive features. In this paper, we answer the question in the affirmative: we show that if one measures fairness using the *mean-difference score*, and sensitive features are subject to noise from the *mutually contaminated learning* model, then owing to a simple identity we only need to change the desired fairness-tolerance. The requisite tolerance can be estimated by leveraging existing noise-rate estimators from the label noise literature. We finally show that our procedure is empirically effective on two case-studies involving sensitive feature censoring.

## 1 Introduction

Classification is concerned with maximally discriminating between a number of pre-defined groups. *Fairness-aware* classification concerns the analysis and design of classifiers that do not discriminate with respect to some sensitive feature (e.g., race, gender, age, income). Recently, much progress has been made on devising appropriate measures of fairness (Calders et al., 2009; Dwork et al., 2011; Feldman, 2015; Hardt et al., 2016; Zafar et al., 2017b,a; Kusner et al., 2017; Kim et al., 2018; Speicher et al., 2018; Heidari et al., 2019), and means of achieving them (Zemel et al., 2013; Zafar et al., 2017b; Calmon et al., 2017; Dwork et al., 2018; Agarwal et al., 2018; Donini et al., 2018; Cotter et al., 2018; Williamson & Menon, 2019; Mohri et al., 2019).

Typically, fairness is achieved by adding constraints which depend on the sensitive feature, and then correcting one's learning procedure to achieve these fairness constraints. For example, suppose the data comprises of pairs of individuals and their loan repay status, and the sensitive feature is gender. Then, we may add a constraint that we should predict equal loan repayment for both men and women (see §3.2 for a more precise statement). However, this and similar approaches assume that we are able to correctly measure or obtain the sensitive feature. In many real-world cases, one may only observe noisy versions of the sensitive feature. For example, survey respondents may choose to conceal or obfuscate their group identity out of concerns of potential mistreatment or outright discrimination.

One is then brought to ask whether fair classification in the presence of such *noisy* sensitive features is still possible. Indeed, if the noise is high enough and all original information about the sensitive

---

[*]Equal contribution

features is lost, then it is as if the sensitive feature was not provided. Standard learners can then be unfair on such data (Dwork et al., 2011; Pedreshi et al., 2008). Recently, Hashimoto et al. (2018) showed that progress is possible, albeit for specific fairness measures. The question of what can be done under a smaller amount of noise is thus both interesting and non-trivial.

In this paper, we consider two practical scenarios where we may only observe noisy sensitive features:

(1) suppose we are releasing data involving human participants. Even if noise-free sensitive features are available, we may wish to *add* noise so as to obfuscate sensitive attributes, and thus protect participant data from potential misuse. Thus, being able to learn fair classifiers under sensitive feature noise is a way to achieve both privacy *and* fairness.

(2) suppose we wish to analyse data where the presence of the sensitive feature is only known for a subset of individuals, while for others the feature value is unknown. For example, patients filling out a form may feel comfortable disclosing that they do not have a pre-existing medical condition; however, some who do have this condition may wish to refrain from responding. This can be seen as a variant of the *positive and unlabelled* (PU) setting (Denis, 1998), where the sensitive feature is present (positive) for some individuals, but absent (unlabelled) for others.

By considering popular measures of fairness and a general model of noise, we show that fair classification is possible under many settings, including the above. Our precise contributions are:

(**C1**) we show that if the sensitive features are subject to noise as per the *mutually contaminated learning model* (Scott et al., 2013a), and one measures fairness using the *mean-difference score* (Calders & Verwer, 2010), then a simple identity (Theorem 2) yields that we only need to change the desired fairness-tolerance. The requisite tolerance can be estimated by leveraging existing noise-rate estimators from the label noise literature, yielding a reduction (Algorithm 1) to regular noiseless fair classification.

(**C2**) we show that our procedure is empirically effective on both case-studies mentioned above.

In what follows, we review the existing literature on learning fair and noise-tolerant classifiers in §2, and introduce the novel problem formulation of noise-tolerant fair learning in §3. We then detail how to address this problem in §4, and empirically confirm the efficacy of our approach in §5.

## 2 Background and related work

We review relevant literature on fair and noise-tolerant machine learning.

### 2.1 Fair machine learning

Algorithmic fairness has gained significant attention recently because of the undesirable social impact caused by bias in machine learning algorithms (Angwin et al., 2016; Buolamwini & Gebru, 2018; Lahoti et al., 2018). There are two central objectives: designing appropriate application-specific fairness criterion, and developing predictors that respect the chosen fairness conditions.

Broadly, fairness objectives can be categorised into individual- and group-level fairness. Individual-level fairness (Dwork et al., 2011; Kusner et al., 2017; Kim et al., 2018) requires the treatment of "similar" individuals to be similar. Group-level fairness asks the treatment of the groups divided based on some sensitive attributes (e.g., gender, race) to be similar. Popular notions of group-level fairness include demographic parity (Calders et al., 2009) and equality of opportunity (Hardt et al., 2016); see §3.2 for formal definitions.

Group-level fairness criteria have been the subject of significant algorithmic design and analysis, and are achieved in three possible ways:

– pre-processing methods (Zemel et al., 2013; Louizos et al., 2015; Lum & Johndrow, 2016; Johndrow & Lum, 2017; Calmon et al., 2017; del Barrio et al., 2018; Adler et al., 2018), which usually find a new representation of the data where the bias with respect to the sensitive feature is explicitly removed.

– methods enforcing fairness during training (Calders et al., 2009; Woodworth et al., 2017; Zafar et al., 2017b; Agarwal et al., 2018), which usually add a constraint that is a proxy of the fairness criteria or add a regularization term to penalise fairness violation.

- post-processing methods (Feldman, 2015; Hardt et al., 2016), which usually apply a thresholding function to make the prediction satisfying the chosen fairness notion across groups.

## 2.2 Noise-tolerant classification

Designing noise-tolerant classifiers is a classic topic of study, concerned with the setting where one's training labels are corrupted in some manner. Typically, works in this area postulate a particular model of label noise, and study the viability of learning under this model. Class-conditional noise (CCN) (Angluin & Laird, 1988) is one such effective noise model. Here, samples from each class have their labels flipped with some constant (but class-specific) probability. Algorithms that deal with CCN corruption have been well studied (Natarajan et al., 2013; Liu & Tao, 2016; Northcutt et al., 2017). These methods typically first estimate the noise rates, which are then used for prediction. A special case of CCN learning is learning from positive and unlabelled data (PU learning) (Elkan & Noto, 2008), where in lieu of explicit negative samples, one has a pool of unlabelled data.

Our interest in this paper will be the *mutually contaminated* (MC) *learning* noise model (Scott et al., 2013a). This model (described in detail in §3.3) captures both CCN and PU learning as special cases (Scott et al., 2013b; Menon et al., 2015), as well as other interesting noise models.

## 3 Background and notation

We recall the settings of standard and fairness-aware binary classification[2], and establish notation.

Our notation is summarized in Table 1.

### 3.1 Standard binary classification

Binary classification concerns predicting the label or *target feature* $Y \in \{0, 1\}$ that best corresponds to a given instance $X \in \mathcal{X}$. Formally, suppose $D$ is a distribution over (instance, target feature) pairs from $\mathcal{X} \times \{0, 1\}$. Let $f \colon \mathcal{X} \to \mathbb{R}$ be a score function, and $\mathcal{F} \subset \mathbb{R}^{\mathcal{X}}$ be a user-defined class of such score functions. Finally, let $\ell : \mathbb{R} \times \{0, 1\} \to \mathbb{R}_+$ be a loss function measuring the disagreement between a given score and binary label. The goal of binary classification is to minimise

$$L_D(f) := \mathbb{E}_{(X,Y) \sim D}[\ell(f(X), Y)]. \tag{1}$$

### 3.2 Fairness-aware classification

In fairness-aware classification, the goal of accurately predicting the target feature $Y$ remains. However, there is an additional *sensitive feature* $A \in \{0, 1\}$ upon which we do not wish to discriminate. Intuitively, some user-defined fairness loss should be roughly the same regardless of $A$.

Formally, suppose $D$ is a distribution over (instance, sensitive feature, target feature) triplets from $\mathcal{X} \times \{0, 1\} \times \{0, 1\}$. The goal of *fairness-aware* binary classification is to find[3]

$$f^* := \underset{f \in \mathcal{F}}{\arg\min}\, L_D(f), \text{ such that } \Lambda_D(f) \leq \tau \tag{2}$$
$$L_D(f) := \mathbb{E}_{(X,A,Y) \sim D}[\ell(f(X), Y)],$$

for user-specified *fairness tolerance* $\tau \geq 0$, and *fairness constraint* $\Lambda_D \colon \mathcal{F} \to \mathbb{R}_+$. Such constrained optimisation problems can be solved in various ways, e.g., convex relaxations (Donini et al., 2018), alternating minimisation (Zafar et al., 2017b; Cotter et al., 2018), or linearisation (Hardt et al., 2016).

A number of fairness constraints $\Lambda_D(\cdot)$ have been proposed in the literature. We focus on two important and specific choices in this paper, inspired by Donini et al. (2018):

$$\Lambda_D^{\mathrm{DP}}(f) := \left| \bar{L}_{D_{0,\cdot}}(f) - \bar{L}_{D_{1,\cdot}}(f) \right| \tag{3}$$
$$\Lambda_D^{\mathrm{EO}}(f) := \left| \bar{L}_{D_{0,1}}(f) - \bar{L}_{D_{1,1}}(f) \right|, \tag{4}$$

**Table 1:** Glossary of commonly used symbols

| Symbol | Meaning | Symbol | Meaning |
|---|---|---|---|
| $X$ | instance | $D_{\text{corr}}$ | corrupted distribution $D$ |
| $A$ | sensitive feature | $f$ | score function $f : \mathcal{X} \to \mathbb{R}$ |
| $Y$ | target feature | $\ell$ | accuracy loss $\ell : \mathbb{R} \times \{0,1\} \to \mathbb{R}_+$ |
| $D$ | distribution $\mathbb{P}(X, A, Y)$ | $L_D$ | expected accuracy loss on $D$ |
| $D_{a,\cdot}$ | distribution $\mathbb{P}(X, A, Y \mid A = a)$ | $\bar{\ell}$ | fairness loss $\bar{\ell} : \mathbb{R} \times \{0,1\} \to \mathbb{R}_+$ |
| $D_{\cdot,y}$ | distribution $\mathbb{P}(X, A, Y \mid Y = y)$ | $\bar{L}_D$ | expected fairness loss on $D$ |
| $D_{a,y}$ | distribution $\mathbb{P}(X, A, Y \mid A = a, Y = y)$ | $\Lambda_D$ | fairness constraint |

where we denote by $D_{a,\cdot}, D_{\cdot,y}$, and $D_{a,y}$ the distributions over $\mathcal{X} \times \{0,1\} \times \{0,1\}$ given by $D_{|A=a}, D_{|Y=y}$, and $D_{|A=a,Y=y}$ and $\bar{\ell} : \mathbb{R} \times \{0,1\} \to \mathbb{R}_+$ is the user-defined fairness loss with corresponding $\bar{L}_D(f) := \mathbb{E}_{(X,A,Y)\sim D}[\bar{\ell}(f(X), Y)]$. Intuitively, these measure the difference in the average of the fairness loss incurred among the instances with and without the sensitive feature.

Concretely, if $\bar{\ell}$ is taken to be $\bar{\ell}(s, y) = \mathbb{1}[\text{sign}(s) \neq 1]$ and the 0-1 loss $\bar{\ell}(s, y) = \mathbb{1}[\text{sign}(s) \neq y]$ respectively, then for $\tau = 0$, (3) and (4) correspond to the *demographic parity* (Dwork et al., 2011) and *equality of opportunity* (Hardt et al., 2016) constraints. Thus, we denote these two relaxed fairness measures *disparity of demographic parity* (DDP) and *disparity of equality of opportunity* (DEO). These quantities are also known as the *mean difference score* in Calders & Verwer (2010).[4]

### 3.3 Mutually contaminated learning

In the framework of learning from mutually contaminated distributions (MC learning) (Scott et al., 2013b), instead of observing samples from the "true" (or "clean") joint distribution $D$, one observes samples from a corrupted distribution $D_{\text{corr}}$. The corruption is such that the observed *class-conditional* distributions are mixtures of their true counterparts. More precisely, let $D_y$ denote the conditional distribution for label $y$. Then, one assumes that

$$
\begin{aligned}
D_{1,\text{corr}} &= (1 - \alpha) \cdot D_1 + \alpha \cdot D_0 \\
D_{0,\text{corr}} &= \beta \cdot D_1 + (1 - \beta) \cdot D_0,
\end{aligned}
\tag{5}
$$

where $\alpha, \beta \in (0, 1)$ are (typically unknown) noise parameters with $\alpha + \beta < 1$.[5] Further, the corrupted base rate $\pi_{\text{corr}} := \mathbb{P}[Y_{\text{corr}} = 1]$ may be arbitrary. The MC learning framework subsumes CCN and PU learning (Scott et al., 2013b; Menon et al., 2015), which are prominent noise models that have seen sustained study in recent years (Jain et al., 2017; Kiryo et al., 2017; van Rooyen & Williamson, 2018; Katz-Samuels et al., 2019; Charoenphakdee et al., 2019).

## 4 Fairness under sensitive attribute noise

The standard fairness-aware learning problem assumes we have access to the true sensitive attribute, so that we can both measure and control our classifier's unfairness as measured by, e.g., Equation 3. Now suppose that rather than being given the sensitive attribute, we get a noisy version of it. We will show that the fairness constraint on the clean distribution is *equivalent* to a *scaled* constraint on the noisy distribution. This gives a simple reduction from fair machine learning in the presence of noise to the regular fair machine learning, which can be done in a variety of ways as discussed in §2.1.

### 4.1 Sensitive attribute noise model

As previously discussed, we use MC learning as our noise model, as this captures both CCN and PU learning as special cases; hence, we automatically obtain results for both these interesting settings.

Our specific formulation of MC learning noise on the sensitive feature is as follows. Recall from §3.2 that $D$ is a distribution over $\mathcal{X} \times \{0,1\} \times \{0,1\}$. Following (5), for unknown noise parameters $\alpha, \beta \in (0,1)$ with $\alpha + \beta < 1$, we assume that the corrupted class-conditional distributions are:

$$
\begin{aligned}
D_{1,\cdot,\text{corr}} &= (1-\alpha) \cdot D_{1,\cdot} + \alpha \cdot D_{0,\cdot} \\
D_{0,\cdot,\text{corr}} &= \beta \cdot D_{1,\cdot} + (1-\beta) \cdot D_{0,\cdot},
\end{aligned}
\tag{6}
$$

and that the corrupted base rate is $\pi_{a,\text{corr}}$ (we write the original base rate, $\mathbb{P}_{(X,A,Y)\sim D}[A=1]$ as $\pi_a$). That is, the distribution over (instance, label) pairs for the group with $A=1$, i.e. $\mathbb{P}(X,Y \mid A=1)$, is assumed to be mixed with the distribution for the group with $A=0$, and vice-versa.

Now, when interested in the EO constraint, it can be simpler to assume that the noise instead satisfies

$$
\begin{aligned}
D_{1,1,\text{corr}} &= (1-\alpha') \cdot D_{1,1} + \alpha' \cdot D_{0,1} \\
D_{0,1,\text{corr}} &= \beta' \cdot D_{1,1} + (1-\beta') \cdot D_{0,1},
\end{aligned}
\tag{7}
$$

for noise parameters $\alpha', \beta' \in (0,1)$. As shown by the following, this is not a different assumption.

**Lemma 1.** *Suppose there is noise in the sensitive attribute only, as given in Equation (6). Then, there exists constants $\alpha', \beta'$ such that Equation (7) holds.*

Although the lemma gives a way to calculate $\alpha', \beta'$ from $\alpha, \beta$, in practice it may be useful to consider (7) independently. Indeed, when one is interested in the EO constraints we will show below that only knowledge of $\alpha', \beta'$ is required. It is often much easier to estimate $\alpha', \beta'$ directly (which can be done in the same way as estimating $\alpha, \beta$ simply by considering $D_{\cdot,1,\text{corr}}$ rather than $D_{\text{corr}}$).

## 4.2 Fairness constraints under MC learning

We now show that the previously introduced fairness constraints for demographic parity and equality of opportunity are automatically robust to MC learning noise in $A$.

**Theorem 2.** *Assume that we have noise as per Equation (6). Then, for any $\tau > 0$ and $f \colon X \to \mathbb{R}$,*

$$
\begin{aligned}
\Lambda_D^{\text{DP}}(f) \leq \tau &\iff \Lambda_{D_{\text{corr}}}^{\text{DP}}(f) \leq \tau \cdot (1-\alpha-\beta) \\
\Lambda_{D_{\cdot,1}}^{\text{EO}}(f) \leq \tau &\iff \Lambda_{D_{\text{corr},\cdot,1}}^{\text{EO}}(f) \leq \tau \cdot (1-\alpha'-\beta'),
\end{aligned}
$$

*where $\alpha'$ and $\beta'$ are as per Equation (7) and Lemma 1.*

The above can be seen as a consequence of the immunity of the *balanced error* (Chan & Stolfo, 1998; Brodersen et al., 2010; Menon et al., 2013) to corruption under the MC model. Specifically, consider a distribution $D$ over an input space $\mathcal{Z}$ and label space $\mathcal{W} = \{0,1\}$. Define

$$
B_D := \mathbb{E}_{Z|W=0}[h_0(Z)] + \mathbb{E}_{Z|W=1}[h_1(Z)]
$$

for functions $h_0, h_1 \colon \mathcal{Z} \to \mathbb{R}$. Then, if for every $z \in \mathbb{R}$ $h_0(z) + h_1(z) = 0$, we have (van Rooyen, 2015, Theorem 4.16), (Blum & Mitchell, 1998; Zhang & Lee, 2008; Menon et al., 2015)

$$
B_{D_{\text{corr}}} = (1-\alpha-\beta) \cdot B_D,
\tag{8}
$$

where $D_{\text{corr}}$ refers to a corrupted version of $D$ under MC learning with noise parameters $\alpha, \beta$. That is, the effect of MC noise on $B_D$ is simply to perform a scaling. Observe that $B_D = \bar{L}_D(f)$ if we set $Z$ to $X \times Y$, $W$ to the sensitive feature $A$, and $h_0((x,y)) = +\bar{\ell}(y, f(x)), h_1((x,y)) = -\bar{\ell}(y, f(x))$. Thus, (8) implies $\bar{L}_D(f) = (1-\alpha-\beta) \cdot \bar{L}_{D_{\text{corr}}}(f)$, and thus Theorem 2.

## 4.3 Algorithmic implications

Theorem 2 has an important algorithmic implication. Suppose we pick a fairness constraint $\Lambda_D$ and seek to solve Equation 2 for a given tolerance $\tau \geq 0$. Then, given samples from $D_{\text{corr}}$, it suffices to simply change the tolerance to $\tau' = \tau \cdot (1-\alpha-\beta)$.

Unsurprisingly, $\tau'$ depends on the noise parameters $\alpha, \beta$. In practice, these will be unknown; however, there have been several algorithms proposed to estimate these from noisy data alone (Scott et al., 2013b; Menon et al., 2015; Liu & Tao, 2016; Ramaswamy et al., 2016; Northcutt et al., 2017). Thus, we may use these to construct estimates of $\alpha, \beta$, and plug these in to construct an estimate of $\tau'$.

In sum, we may tackle fair classification in the presence of noisy $A$ by suitably combining *any* existing fair classification method (that takes in a parameter $\tau$ that is proportional to mean-difference score of some fairness measures), and *any* existing noise estimation procedure. This is summarised in Algorithm 1. Here, FairAlg is any existing fairness-aware classification method that solves Equation 2, and NoiseEst is any noise estimation method that estimates $\alpha, \beta$.

---

**Algorithm 1** Reduction-based algorithm for fair classification given noisy $A$.

---

**Input:** Training set $S = \{(x_i, y_i, a_i)\}_{i=1}^n$, scorer class $\mathcal{F}$, fairness tolerance $\tau \geq 0$, fairness constraint $\Lambda(\cdot)$, fair classification algorithm FairAlg, noise estimation algorithm NoiseEst
**Output:** Fair classifier $f^* \in \mathcal{F}$
 1: $\hat{\alpha}, \hat{\beta} \leftarrow$ NoiseEst$(S)$
 2: $\tau' \leftarrow (1 - \hat{\alpha} - \hat{\beta}) \cdot \tau$
 3: **return** FairAlg$(S, \mathcal{F}, \Lambda, \tau')$

---

### 4.4 Noise rate and sample complexity

So far, we have shown that at a distribution level, fairness with respect to the noisy sensitive attribute is equivalent to fairness with respect to the real sensitive attribute. However, from a sample complexity perspective, a higher noise rate will require a larger number of samples for the empirical fairness to generalize well, i.e., guarantee fairness at a distribution level. A concurrent work by Mozannar et al. (2019) derives precise sample complexity bounds and makes this relationship explicit.

### 4.5 Connection to privacy and fairness

While Algorithm 1 gives a way of achieving fair classification on an already noisy dataset such as the use case described in example (2) of §1, it can also be used to simultaneously achieve fairness and privacy. As described in example (1) of §1, the very nature of the sensitive attribute makes it likely that even if noiseless sensitive attributes are available, one might want to add noise to guarantee some form of privacy. Note that simply removing the feature does not suffice, because it would make difficult the task of developing fairness-aware classifiers for the dataset (Gupta et al., 2018). Formally, we can give the following privacy guarantee by adding CCN noise to the sensitive attribute.

**Lemma 3.** *To achieve $(\epsilon, \delta = 0)$ differential privacy on the sensitive attribute we can add CCN noise with $\rho^+ = \rho^- = \rho \geq \frac{1}{\exp(\epsilon)+1}$ to the sensitive attribute.*

Thus, if a desired level of differential privacy is required before releasing a dataset, one could simply add the required amount of CCN noise to the sensitive attributes, publish this modified dataset as well as the noise level, and researchers could use Algorithm 1 (without even needing to estimate the noise rate) to do fair classification as usual.

There is previous work that tries to preserve privacy of individuals' sensitive attributes while learning a fair classifier. Kilbertus et al. (2018) employs the cryptographic tool of secure multiparty computation (MPC) to try to achieve this goal. However, as noted by Jagielski et al. (2018), the individual information that MPC tries to protect can still be inferred from the learned model. Further, the method of Kilbertus et al. (2018) is limited to using demographic parity as the fairness criteria.

A more recent work of Jagielski et al. (2018) explored preserving differential privacy (Dwork, 2006) while maintaining fairness constraints. The authors proposed two methods: one adds Laplace noise to training data and apply the post-processing method in Hardt et al. (2016), while the other modifies the method in Agarwal et al. (2018) using the exponential mechanism as well as Laplace noise. Our work differs from them in three major ways:

- (*1*) our work allows for fair classification to be done using *any in-process* fairness-aware classifier that allows user to specify desired fairness level. On the other hand, the first method of Jagielski et al. (2018) requires the use of a post-processing algorithm (which generally have worse trade-offs than in-processing algorithms Agarwal et al. (2018)), while the second method requires the use of a single *particular* classifier.
- (*2*) our focus is on designing fair-classifiers with noise-corrupted sensitive attributes; by contrast, the main concern in Jagielski et al. (2018) is achieving differential privacyand thus they do not discuss how to deal with noise that is already present in the dataset.

(*3*) our method is shown to work for a large class of different fairness definitions.

Finally, a concurrent work of Mozannar et al. (2019) builds upon our method for the problem of preserving privacy of the sensitive attribute. The authors use a randomized response procedure on the sensitive attribute values, followed by a two-step procedure to train a fair classifier using the processed data. Theoretically, their method improves upon the sample complexity of our method and extends our privacy result to the case of non-binary groups. However, their method solely focuses on preserving privacy rather than the general problem of sensitive attribute noise.

## 5 Experiments

We demonstrate that it is viable to learn fair classifiers given noisy sensitive features.[6] As our underlying fairness-aware classifier, we use a modified version of the classifier implemented in Agarwal et al. (2018) with the DDP and DEO constraints which, as discussed in §3.2, are special cases of our more general constraints (3) and (4). The classifier's original constraints can also be shown to be noise-invariant but in a slightly different way (see Appendix C for a discussion). An advantage of this classifier is that it is shown to reach levels of fairness violation that are very close to the desired level ($\tau$), i.e., for small enough values of $\tau$ it will reach the constraint boundary.

While we had to choose a particular classifier, our method can be used before using any downstream fair classifier as long as it can take in a parameter $\tau$ that controls the strictness of the fairness constraint and that its constraints are special cases of our very general constraints (3) and (4).

### 5.1 Noise setting

Our case studies focus on two common special cases of MC learning: CCN and PU learning. Under CCN noise the sensitive feature's value is randomly flipped with probability $\rho^+$ if its value was 1, or with probability $\rho^-$ if its value was 0. As shown in Menon et al. (2015, Appendix C), CCN noise is a special case of MC learning. For PU learning we consider the censoring setting (Elkan & Noto, 2008) which is a special case of CCN learning where one of $\rho^+$ and $\rho^-$ is 0. While our results also apply to the case-controlled setting of PU learning (Ward et al., 2009), the former setting is more natural in our context. Note that from $\rho^+$ and $\rho^-$ one can obtain $\alpha$ and $\beta$ as described in Menon et al. (2015).

### 5.2 Benchmarks

For each case study, we evaluate our method (termed cor scale); recall this scales the input parameter $\tau$ using Theorem 2 and the values of $\rho^+$ and $\rho^-$, and then uses the fair classifier to perform classification. We compare our method with three different baselines. The first two trivial baselines are applying the fair classifier directly on the non-corrupted data (termed nocor) and on the corrupted data (termed cor). While the first baseline is clearly the ideal, it won't be possible when only the corrupted data is available. The second baseline should show that there is indeed an empirical need to deal with the noise in some way and that it cannot simply be ignored.

The third, non-trivial, baseline (termed denoise) is to first denoise $A$ and then apply the fair classifier on the denoised distribution. This denoising is done by applying the RankPrune method in Northcutt et al. (2017). Note that we provide RankPrune with the same known values of $\rho^+$ and $\rho^-$ that we use to apply our scaling so this is a fair comparison to our method. Compared to denoise, we do *not* explicitly infer individual sensitive feature values; thus, our method does not compromise privacy.

For both case studies, we study the relationship between the input parameter $\tau$ and the testing error and fairness violation. For simplicity, we only consider the DP constraint.

### 5.3 Case study: privacy preservation

In this case study, we look at COMPAS, a dataset from Propublica (Angwin et al., 2016) that is widely used in the study of fair algorithms. Given various features about convicted individuals, the task is to predict recidivism and the sensitive attribute is race. The data comprises 7918 examples and 10 features. In our experiment, we assume that to preserve differential privacy, CCN noise with $\rho^+ = \rho^- = 0.15$ is added to the sensitive attribute. As per Lemma 3, this guarantees ($\epsilon, \delta = 0$)

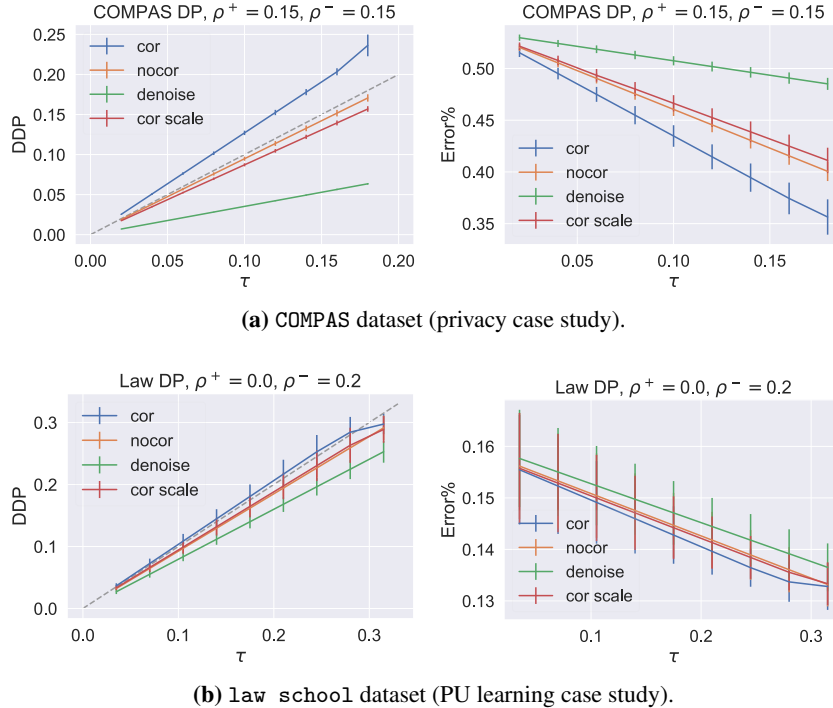

**(a)** `COMPAS` dataset (privacy case study).

**(b)** `law school` dataset (PU learning case study).

**Figure 1:** Relationship between input fairness tolerance $\tau$ versus DP fairness violation (left panels), and versus error (right panels). Our method (cor scale) achieves approximately the ideal fairness violation (indicated by the gray dashed line in the left panels), with only a mild degradation in accuracy compared to training on the uncorrupted data (indicated by the nocor method). Baselines that perform no noise-correction (cor) and explicitly denoise the data (denoise) offer suboptimal tradeoffs by comparison; for example, the former achieves slightly lower error rates, but does so at the expense of greater fairness violation.

differential privacy with $\epsilon = 1.73$. We assume that the noise level $\rho$ is released with the dataset (and is thus known). We performed fair classification on this noisy data using our method and compare the results to the three benchmarks described above.

Figure 1a shows the average result over three runs each with a random 80-20 training-testing split. (Note that fairness violations and errors are calculated with respect to the true uncorrupted features.) We draw two key insights from this graph:

(i) in terms of fairness violation, our method (cor scale) approximately achieves the desired fairness tolerance (shown by the gray dashed line). This is both expected and ideal, and it matches what happens when there is no noise (nocor). By contrast, the naïve method cor strongly violates the fairness constraint.

(ii) in terms of accuracy, our method only suffers mildly compared with the ideal noiseless method (nocor); some degradation is expected as noise will lead to some loss of information. By contrast, denoise sacrifices much more predictive accuracy than our method.

In light of both the above, our method is seen to achieve the best overall tradeoff between fairness and accuracy. Experimental results with EO constraints, and other commonly studied datasets in the fairness literature (adult, german), show similar trends as in Figure 1a, and are included in Appendix D for completeness.

## 5.4 Case study: PU learning

In this case study, we consider the dataset `law school`, which is a subset of the original dataset from LSAC (Wightman, 1998). In this dataset, one is provided with information about various individuals (grades, part time/full time status, age, etc.) and must determine whether or not the individual passed the bar exam. The sensitive feature is race; we only consider black and white. After prepossessing

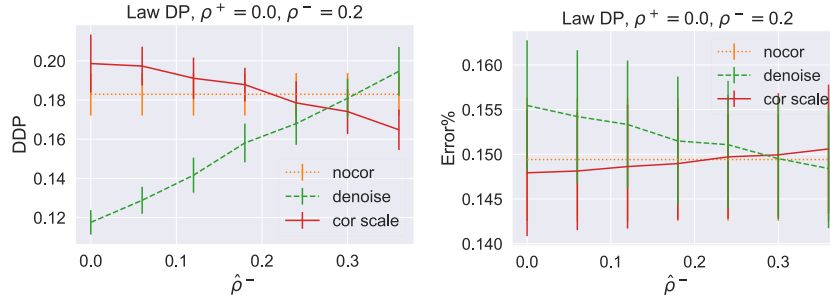

**Figure 2:** Relationship between the estimated noise level $\hat{\rho}^-$ and fairness violation/error on the `law school` dataset using DP constraint (testing curves), with $\hat{\rho}^+ = 0$ and $\tau = 0.2$. Our method (cor scale) is not overly sensitive to imperfect estimates of the noise rate, evidenced by its fairness violation and accuracy closely tracking that of training on the uncorrupted data (nocor) as $\hat{\rho}^-$ is varied. That is, red curve in the left plot closely tracks the yellow reference curve. By contrast, the baseline that explicitly denoises the data (denoise) deviates strongly from nocor, and is sensitive to small changes in $\hat{\rho}^-$. This illustrates that our method performs well even when noise rates must be estimated.

the data by removing instances that had missing values and those belonging to other ethnicity groups (neither black nor white) we were left with 3738 examples each with 11 features.

While the data ostensibly provides the true values of the sensitive attribute, one may imagine having access to only PU information. Indeed, when the data is collected one could imagine that individuals from the minority group would have a much greater incentive to conceal their group membership due to fear of discrimination. Thus, any individual identified as belonging to the majority group could be assumed to have been correctly identified (and would be part of the positive instances). On the other hand, no definitive conclusions could be drawn about individuals identified as belonging to the minority group (these would therefore be part of the unlabelled instances).

To model a PU learning scenario, we added CCN noise to the dataset with $\rho^+ = 0$ and $\rho^- = 0.2$. We initially assume that the noise rate is known. Figure 1b shows the average result over three runs under this setting each with a random 80-20 training-testing split. We draw the same conclusion as before: our method achieves the highest accuracy while respecting the specified fairness constraint.

Unlike in the privacy case, the noise rate in the PU learning scenario is usually unknown in practice, and must be estimated. Such estimates will inevitably be approximate. We thus evaluate the impact of the error of the noise rate estimate on all methods. In Figure 2, we consider a PU scenario where we only have access to an estimate $\hat{\rho}^-$ of the negative noise rate, whose true value is $\rho^- = 0.2$. Figure 2 shows the impact of different values of $\hat{\rho}^-$ on the fairness violation and error. We see that that as long as this estimate is reasonably accurate, our method performs the best in terms of being closest to the case of running the fair algorithm on uncorrupted data.

In sum, these results are consistent with our derivation and show that our method cor scale can achieve the desired degree of fairness while minimising loss of accuracy. Appendix E includes results for different settings of $\tau$, noise level, and on other datasets showing similar trends.

## 6    Conclusion and future work

In this paper, we showed both theoretically and empirically that even under the very general MC learning noise model (Scott et al., 2013a) on the sensitive feature, fairness can still be preserved by scaling the input unfairness tolerance parameter $\tau$. In future work, it would be interesting to consider the case of categorical sensitive attributes (as applicable, e.g., for race), and the more challenging case of instance-dependent noise (Awasthi et al., 2015). We remark also that in independent work, Awasthi et al. (2019) studied the effect of sensitive attribute noise on the post-processing method of Hardt et al. (2016). In particular, they identified conditions when such post-processing can still yield an approximately fair classifier. Our approach has an advantage of being applicable to a generic in-processing fair classifier; however, their approach also handles the case where the sensitive feature is used as an input to the classifier. Exploration of the synthesis of the two approaches is another promising direction for future work.

## Footnotes

[2]For simplicity, we consider the setting of binary target and sensitive features. However, our derivation and method can be easily extended to the multi-class setting.

[3]Here, $f$ is assumed to not be allowed to use $A$ at test time, which is a common legal restriction (Lipton et al., 2018). Of course, $A$ can be used at training time to find an $f$ which satisfies the constraint.

[4]Keeping $\bar{\ell}$ generic allows us to capture a range of group fairness definitions, not just demographic parity and equality of opportunity; e.g., disparate mistreatment (Zafar et al., 2017b) corresponds to using the 0-1 loss and $\Lambda_D^{\text{DP}}$, and equalized odds can be captured by simply adding another constraint for $Y = 0$ along with $\Lambda_D^{\text{EO}}$.

[5]The constraint imposes no loss of generality: when $\alpha + \beta > 1$, we can simply flip the two labels and apply our theorem. When $\alpha + \beta = 1$, all information about the sensitive attribute is lost. This pathological case is equivalent to not measuring the sensitive attribute at all.

[6]Source code is available at https://github.com/AIasd/noise_fairlearn.

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
