[Supplementary Material 1]

# Noise-tolerant fair classification

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

# Supplementary material for "Noise-tolerant fair classification"

## A Proofs of results in the main body

### A.1 Proof of Lemma 1

*Proof.* Suppose that we have noise as given by Equation (5). We denote by $A$ the random variable denoting the value of the true sensitive attribute and by $A_{\mathrm{corr}}$ the random variable denoting the value of the corrupted sensitive attribute.

Then, for any measurable subset of instances $U$,

$$
\begin{aligned}
&\mathbb{P}[X \in U \mid Y = 1, A_{\mathrm{corr}} = 1] \\
&= \frac{\mathbb{P}[X \in U, Y = 1 \mid A_{\mathrm{corr}} = 1]}{\mathbb{P}[Y = 1 \mid A_{\mathrm{corr}} = 1]} \\
&= \frac{\mathbb{P}[X \in U, Y = 1 \mid A_{\mathrm{corr}} = 1]}{(1 - \alpha)\mathbb{P}[Y = 1 \mid A = 1] + \alpha\mathbb{P}[Y = 1 \mid A = 0]} \\
&= \frac{(1 - \alpha)\mathbb{P}[X \in U, Y = 1 \mid A = 1]}{(1 - \alpha)\mathbb{P}[Y = 1 \mid A = 1] + \alpha\mathbb{P}[Y = 1 \mid A = 0]} \\
&\quad + \frac{\alpha\mathbb{P}[X \in U, Y = 1 \mid A = 0]}{(1 - \alpha)\mathbb{P}[Y = 1 \mid A = 1] + \alpha\mathbb{P}[Y = 1 \mid A = 0]} \\
&= \frac{(1 - \alpha)\mathbb{P}[Y = 1 \mid A = 1]\mathbb{P}[X \in U \mid Y = 1, A = 1]}{(1 - \alpha)\mathbb{P}[Y = 1 \mid A = 1] + \alpha\mathbb{P}[Y = 1 \mid A = 0]} \\
&\quad + \frac{\alpha\mathbb{P}[Y = 1 \mid A = 0]\mathbb{P}[X \in U \mid Y = 1, A = 0]}{(1 - \alpha)\mathbb{P}[Y = 1 \mid A = 1] + \alpha\mathbb{P}[Y = 1 \mid A = 0]} \\
&= (1 - \alpha')\mathbb{P}[X \in U \mid Y = 1, A = 1] \\
&\quad + \alpha'\mathbb{P}[X \in U \mid Y = 1, A = 0],
\end{aligned}
$$

where in the last equality we set

$$
\alpha' := \frac{\alpha\mathbb{P}[Y = 1 \mid A = 0]}{(1 - \alpha)\mathbb{P}[Y = 1 \mid A = 1] + \alpha\mathbb{P}[Y = 1 \mid A = 0]}.
$$

Note that the last equality is equivalent to the first equality of Equation (6) with $\alpha'$ as in the lemma.

The proof for $\beta'$ is exactly the same and simply expands $\mathbb{P}[X \in U \mid Y = 1, A_{\mathrm{corr}} = 0]$ instead of $\mathbb{P}[X \in U \mid Y = 1, A_{\mathrm{corr}} = 1]$. $\qquad\square$

### A.2 Proof of Theorem 2

*Proof.* For the DP-like constraints simply note that by definition of $D_{\mathrm{corr}}$ we have that

$$
\bar{L}_{D_{0,\cdot,\mathrm{corr}}}(f) = (1 - \beta) \cdot \bar{L}_{D_{0,\cdot}}(f) + \beta \cdot \bar{L}_{D_{1,\cdot}}(f)
$$

and similarly,

$$
\bar{L}_{D_{1,\cdot,\mathrm{corr}}}(f) = (1 - \alpha) \cdot \bar{L}_{D_{1,\cdot}}(f) + \alpha \cdot \bar{L}_{D_{0,\cdot}}(f)
$$

Thus we have that

$$
\begin{aligned}
\bar{L}_{D_{0,\cdot,\mathrm{corr}}}(f) - \bar{L}_{D_{1,\cdot,\mathrm{corr}}}(f) = (1 - \alpha - \beta)\cdot \\
(\bar{L}_{D_{0,\cdot}}(f) - \bar{L}_{D_{1,\cdot}}(f)),
\end{aligned}
$$

which immediately implies the desired result.

The result for the EO constraint is obtained in the exact same way by simply replacing $D_{a,\cdot}$ with $D_{a,1}$, $D_{a,\cdot,\mathrm{corr}}$ with $D_{a,1,\mathrm{corr}}$, and $\alpha$ and $\beta$ with $\alpha'$ and $\beta'$. $\qquad\square$

## A.3 Proof of Lemma 3

*Proof.* Basic definitions in Differential Privacy are provided in Appendix B. Consider an instance $\{x_i, y_i, a_i\}$ with only $x_i$ disclosed by an attacker. Assume the sensitive attribute $a_i$ is queried. Denote $\hat{a}_i$ to be the sensitive attribute of the instance after adding noise i.e. being flipped with probability $\rho$. The attacker is interested in knowing if $a_i = 0$ or $a_i = 1$ by querying $\hat{a}_i$.

Since

$$\frac{\mathbb{P}[\hat{a}_i = 1 | a_i = 1]}{\mathbb{P}[\hat{a}_i = 1 | a_i = 0]} = \frac{\mathbb{P}[\hat{a}_i = 0 | a_i = 0]}{\mathbb{P}[\hat{a}_i = 0 | a_i = 1]}$$

we can reason in a similar way for $\hat{a}_i = 0$. Thus, let us focus on the case where $\hat{a}_i = 1$. Let us consider two neighbor instances $\{x_i, y_i, 0\}$ and $\{x_i, y_i, 1\}$. Essentially, we want to upper-bound the ratio

$$\frac{\mathbb{P}[\hat{a} = 1 | a_i = 1]}{\mathbb{P}[\hat{a} = 1 | a_i = 0]} := \frac{1 - \rho}{\rho}$$

by $\exp(\epsilon)$, and lower-bound the ratio by $\exp(-\epsilon)$. The lower bound is always true since $\rho < 0.5$. For the upper-bound, We have:

$$\frac{1 - \rho}{\rho} \leq \exp(\epsilon) \iff \rho \geq \frac{1}{\exp(\epsilon) + 1}.$$

$\square$

## B  Background on Differential Privacy

The following definitions are from Appendix Dwork (2006). They are used for the proof of Lemma 3 in A.

**Probability simplex**: Given a discrete set $B$, the probability simplex over $B$, denoted $\Delta(B)$ is:

$$\Delta(B) = \{x \in \mathbb{R}^{|B|} : \forall i, x_i \geq 0, \text{ and } \sum_{i=1}^{|B|} x_i = 1\}.$$

**Randomized Algorithms**: A randomized algorithm $\mathcal{M}$ with domain $A$ and range $B$ is an algorithm associated with a total map $M : A \to \Delta(B)$. On input $a \in A$, the algorithm $\mathcal{M}$ outputs $\mathcal{M}(a) = b$ with probability $(M(a))_b$ for each $b \in B$. The probability space is over the coin flips of the algorithm $\mathcal{M}$.

For simplicity we will avoid implementation details and we will consider databases as histograms. Given a universe $\mathcal{X}$ an histogram over $\mathcal{X}$ is an object in $\mathbb{N}^{|\mathcal{X}|}$. We can bake in the presence or absence of an individual notion in a definition of distance between databases.

**Distance Between Databases**: The $l_1$ norm $\|x\|_1$ of a database $x \in \mathbb{N}^{|\mathcal{X}|}$ is defined as:

$$\|x\|_1 = \sum_{i=1}^{|\mathcal{X}|} x_i.$$

The $l_1$ distance between two databases $x$ and $y$ is defined as $\|x - y\|_1$.

**Differential Privacy**: A randomized algorithm $\mathcal{M}$ with domain $\mathbb{N}^{|\mathcal{X}|}$ is $(\epsilon, \delta)$-differentially private if for all $S \subseteq \text{Range}(\mathcal{M})$ and for all $x, y \in \mathbb{N}^{\mathcal{X}}$ such that $\|x - y\|_1 \leq 1$:

$$\mathbb{P}[\mathcal{M}(x) \in S] \leq \exp(\epsilon) \cdot \mathbb{P}[\mathcal{M}(y) \in S] + \delta,$$

where the probability space is over the coin flips of the mechanism $\mathcal{M}$.

 # C  Relationship between mean-difference score and the constraint used in
 ## Agarwal et al. (2018)

Agarwal et al. (2018) adopts slightly different fairness constraints than ours. Using our notation and letting $c_f(X) = \text{sign}(f(X))$, instead of bounding $\Lambda_D^{\text{DP}}(f)$ by $\tau$, they bound

$$\max_{a \in \{0,1\}} \left| \mathbb{E}_{D_{a,\cdot}}[c_f(X)] - \mathbb{E}_D[c_f(X)] \right|$$

and

$$\max_{a \in \{0,1\}} \left| \mathbb{E}_{D_{a,1}}[c_f(X)] - \mathbb{E}_{D_{\cdot,1}}[c_f(X)] \right|$$

474 for DP and EO respectively by $\tau$. The two have the following relationship.

475 **Theorem 4.** *Under the setting of fair binary classification with a single binary sensitive attribute*
476 *and using $\bar{\ell}(s,y) = \mathbb{1}[\text{sign}(s)]$ we have that*

$$\max_{a \in \{0,1\}} \left| \mathbb{E}_{D_{a,\cdot}}[c_f(X)] - \mathbb{E}_D[c_f(X)] \right| = \max_{a \in \{0,1\}} (\mathbb{P}[A = 0], \mathbb{P}[A = 1]) \Lambda_D^{\text{DP}}(f)$$

477 *and*

$$\max_{a \in \{0,1\}} \left| \mathbb{E}_{D_{a,1}}[c_f(X)] - \mathbb{E}_{D_{\cdot,1}}[c_f(X)] \right| = \max_{a \in \{0,1\}} (\mathbb{P}[A = 0 \mid Y = 1], \mathbb{P}[A = 1 \mid Y = 1]) \Lambda_D^{\text{EO}}(f)$$

478 *Proof.* For the DP case,

$$\left| \mathbb{E}_{D_{1,\cdot}}[c_f(X)] - \mathbb{E}_D[c_f(X)] \right|$$
$$= \left| \mathbb{E}_{D_{1,\cdot}}[c_f(X)] - (\mathbb{P}[A = 1]\mathbb{E}_{D_{1,\cdot}}[c_f(X)] + \mathbb{P}[A = 0]\mathbb{E}_{D_{0,\cdot}}[c_f(X)]) \right|$$
$$= \left| (1 - \mathbb{P}[A = 1])\mathbb{E}_{D_{1,\cdot}}[c_f(X)] - \mathbb{P}[A = 0]\mathbb{E}_{D_{0,\cdot}}[c_f(X)] \right|$$
$$= \left| \mathbb{P}[A = 0]\mathbb{E}_{D_{1,\cdot}}[c_f(X)] - \mathbb{P}[A = 0]\mathbb{E}_{D_{0,\cdot}}[c_f(X)] \right|$$
$$= \mathbb{P}[A = 0] \left| (\mathbb{E}_{D_{1,\cdot}}[c_f(X)] - \mathbb{E}_{D_{0,\cdot}}[c_f(X)]) \right|$$
$$= \mathbb{P}[A = 0] \left| \bar{L}_{D_{0,\cdot}}(f) - \bar{L}_{D_{1,\cdot}}(f) \right|$$
$$= \mathbb{P}[A = 0]\Lambda_D^{\text{DP}}(f)$$

479 and similarly

$$\left| \mathbb{E}_{D_{0,\cdot}}[c_f(X)] - \mathbb{E}_D[c_f(X)] \right| = \mathbb{P}[A = 1]\Lambda_D^{\text{DP}}(f)$$

480 so the theorem holds.

481 The result for the EO case is proved in exactly the same way by simply replacing $\mathbb{P}[A = 0], \mathbb{P}[A = 1]$,
482 $D_{a,\cdot}$ and $D$ with $\mathbb{P}[A = 0 \mid Y = 1], \mathbb{P}[A = 1 \mid Y = 1], D_{a,1}$ and $D_{\cdot,1}$ respectively.

483 $\qquad\qquad\qquad\qquad\qquad\qquad\qquad\qquad\qquad\qquad\qquad\qquad\qquad\qquad\qquad\qquad\qquad\qquad\qquad\square$

484 We then have the following as an immediate corollary.

485 **Corollary 5.** *Assuming that we have noise as described above by Equation* (5) *and that we*
486 *take $\bar{\ell}(s,y) = \mathbb{1}[\text{sign}(s)]$ then we have that if $\max_{a \in \{0,1\}}(\mathbb{P}_D[A = 0], \mathbb{P}_D[A = 1]) =$*
487 $\max_{a \in \{0,1\}}(\mathbb{P}_{D_{\text{corr}}}[A = 0], \mathbb{P}_{D_{\text{corr}}}[A = 1])$ *then:*

$$\max_{a \in \{0,\cdot\}} \left| \mathbb{E}_{D_{a,\cdot}}[c_f(X)] - \mathbb{E}_D[c_f(X)] \right| < \tau \iff \max_{a \in \{0,1\}} \left| \mathbb{E}_{D_{a,\cdot,corr}}[c_f(X)] - \mathbb{E}_{D_{corr}}[c_f(X)] \right| < \tau \cdot (1 - \alpha - \beta).$$

488 *And if $\max_{a \in \{0,1\}}(\mathbb{P}_{D_{\cdot,1}}[A = 0], \mathbb{P}_{D_{\cdot,1}}[A = 1]) = \max_{a \in \{0,1\}}(\mathbb{P}_{D_{\cdot,1,\text{corr}}}[A = 0], \mathbb{P}_{D_{\cdot,1,\text{corr}}}[A = 1])$*
489 *then:*

$$\max_{a \in \{0,1\}} \left| \mathbb{E}_{D_{a,1}}[c_f(X)] - \mathbb{E}_{D_{\cdot,1}}[c_f(X)] \right| < \tau \iff \max_{a \in \{0,1\}} \left| \mathbb{E}_{D_{a,1,corr}}[c_f(X)] - \mathbb{E}_{D_{\cdot,1,corr}}[c_f(X)] \right| < \tau \cdot (1 - \alpha' - \beta').$$

490 Even if the noise does not satisfy these new assumptions, we can still bound the constraint. Note
491 that both $\max_{a \in \{0,1\}}(\mathbb{P}[A = 0], \mathbb{P}[A = 1])$ and $\max_{a \in \{0,1\}}(\mathbb{P}[A = 0 \mid Y = 1], \mathbb{P}[A = 1 \mid Y = 1])$
492 have values between $0.5$ and $1$. Thus,

$$\frac{1}{2}\Lambda_D^{\text{DP}}(f) \leq \max_{a \in \{0,1\}} \left| \mathbb{E}_{D_{a,\cdot}}[c_f(X)] - \mathbb{E}_D[c_f(X)] \right| \leq \Lambda_D^{\text{DP}}(f)$$

$$\frac{1}{2}\Lambda_D^{\mathrm{EO}}(f) \le \max_{a\in\{0,1\}} \left|\mathbb{E}_{D_{a,1}}[c_f(X)] - \mathbb{E}_{D_{\cdot,1}}[c_f(X)]\right| \le \Lambda_D^{\mathrm{EO}}(f),$$

and therefore the following corollary holds:

**Corollary 6.** *Assuming that we have noise as described above by Equation* (5) *and that we take* $\bar{\ell}(s,y) = \mathbb{1}[\mathrm{sign}(s)]$ *then we have that:*

$$\max_{a\in\{0,1\}} \left|\mathbb{E}_{D_{a,\cdot,corr}}[c_f(X)] - \mathbb{E}_{D_{corr}}[c_f(X)]\right| < \frac{1}{2}\tau\cdot(1-\alpha-\beta) \Rightarrow \max_{a\in\{0,1\}} \left|\mathbb{E}_{D_{a,\cdot}}[c_f(X)] - \mathbb{E}_{D}[c_f(X)]\right| < \tau$$

*and,*

$$\max_{a\in\{0,1\}} \left|\mathbb{E}_{D_{a,1,corr}}[c_f(X)] - \mathbb{E}_{D_{\cdot,1,corr}}[c_f(X)]\right| < \frac{1}{2}\tau\cdot(1-\alpha'-\beta') \Rightarrow \max_{a\in\{0,1\}} \left|\mathbb{E}_{D_{a,1}}[c_f(X)] - \mathbb{E}_{D_{\cdot,1}}[c_f(X)]\right| < \tau.$$

In addition to giving a simple way to use the classifier of Agarwal et al. (2018) without any modifica-
tion, these results seem to indicate that with small modifications our scaling method can apply to an
even wider range of fair classifiers than formally shown.

## D    More results for the privacy case study

In this section we give some additional results for the privacy case study.

Figure 3 shows additional results on `COMPASS` for different noise levels $\rho^+ = \rho^- \in \{0.15, 0.3\}$.

**Figure 3:** Relationship between input $\tau$ and fairness violation/error on the `COMPAS` dataset using DP constraint (testing curves). The gray dashed line represents the ideal fairness violation.

Figure 4 shows the results under the EO constraint for the `COMPAS` dataset. That is, the dataset and setting is the same as described in section 5.3 but with the EO constraint instead of the DP constraint. We see that the trends are the same.

**Figure 4:** (EO)(testing and training) Relationship between input $\tau$ and fairness violation/error on the `COMPAS` dataset.

Figures 5 and Figure 6 show results on the `bank` dataset (Ban) with the DP and EO constraints respectively. This dataset is a subset of the original Bank Marketing dataset from the UCI repository (Dheeru & Karra Taniskidou, 2017). The task is to predict if a client subscribes a term deposit. The sensitive attribute is if a person is middle aged(i.e. has an age between 25 and 60). The data comprises 11162 examples and 17 features. Again we note that the trends are the same.

**Figure 5:** (DP)(testing and training) Relationship between input $\tau$ and fairness violation/error on the `bank` dataset.

**Figure 6:** (EO)(testing and training) Relationship between input $\tau$ and fairness violation/error on the `bank` dataset.

## E  More results for the PU case study

In this section we give some additional results for the PU case study.

Figure 7 shows additional results on `law school` for different noise levels $\rho^+ = \rho^- \in \{0.2, 0.4\}$. Figure 8 shows additional results under noise rate estimation on `law school` for different upper bound of fairness violation: $\tau \in \{0.1, 0.3\}$.

**Figure 7:** Relationship between input $\tau$ and fairness violation/error on the `law school` dataset using DP constraint (testing curves). The gray dashed line represents the ideal fairness violation. Note that in some of the graphs, the red line and the orange line perfectly overlap with each other.

Figure 9 and Figure 10 show the results under PU noise on the `german` dataset, which is another dataset from the UCI repository (Dheeru & Karra Taniskidou, 2017). The task is to predict if one has good credit and the sensitive attribute is whether a person is foreign. The data comprises 1000 examples and 20 features. The trends are similar to those for the `law school` dataset.

**Figure 8:** Relationship between the estimated noise level $\hat{\rho}^-$ and fairness violation/error on the `law school` dataset using DP constraint (testing curves) at $\tau \in \{0.1, 0.3\}$, with $\hat{\rho}^+ = 0$.

**Figure 9:** (DP)(training and testing) Relationship between input $\tau$ and fairness violation/error on the `german` dataset.

**Figure 10:** Relationship between the estimated noise level $\hat{\rho}^+$ and fairness violation/error on the `german` dataset using DP constraint (testing curves). Note that $\hat{\rho}^-$ is fixed to 0 and $\tau = 0.04$.

## F  The influence of different noise levels

Figure 11 explores the influence of the noise level on the trends and relationships between our method's performance and that of the benchmarks. We run these experiments on the UCI `adult` dataset, which is another dataset from the UCI repository (Dheeru & Karra Taniskidou, 2017). The task is to predict if one has income more than 50K and gender is the sensitive attribute. The data comprises 48842 examples and 14 features. We run these experiments with the DP constraint under different CCN noise levels ($\rho^+ = \rho^- \in \{0.01, 0.1, 0.2, 0.3, 0.4, 0.48\}$). We include both training and testing curves for completeness. As we can see, as the noise increases the gap between the corrupted data curves and the uncorrupted data curve increases. It becomes very hard to get close to the non-corrupted case when noise becomes too high.

**Figure 11:** Relationship between input $\tau$ and fairness violation/error on the `adult` dataset for various noise levels. From left to right: testing fairness violation, testing error, training fairness violation, and training error. Different noise levels from top to bottom.

[Supplementary Material 2]

# Noise-tolerant fair classification