[Reviews · NeurIPS 2019]

Reviewer 1



At one level I really like the very cute observation (Theorem 2) presented in the paper and acknowledge that it has potentially interesting implications. On the other hand, I see the following major issues with this work, which makes me feel that it does not rise to the level of a NeurIPS paper. (a) Very limited generalizability: The findings do not generalize to (i) notions of fairness other than demographic parity (equalized odds is nothing but demographic parity over positively labeled data points), (ii) scenarios where there non-binary sensitive features -- e.g., it is unclear how the observation generalizes to scenarios where there are more than 2 races in a population, (iii) scenarios where the noisy labels deviate from mutual contamination model with constraints (\alpha + \beta < 1) (b) Notations and definitions are messed up at different places -- E.g., equations (4) and (5) are inconsistent in the way they define the corrupted distribution. equation (1) does not define \Lambda and seems to redefine a loss function already defined before. The loss function with a bar on top in equation (2) is not defined. (c) Something I did not quite get: In equation (5), \alpha D_{0,.} goes in one way, while (1 - \beta) D_{0,.} goes the other way. But, given that \alpha and \beta don't sum up to 1, what happens to the remaining data points? (d) Technically, the paper does not have a lot to add. But, the primary contribution of the paper is the cute observation and its application scenarios in practice. I wonder if this paper might be better suited at a conference focussed on the topic.

Reviewer 2



I think this is a nice model and I like the clean analysis. The paper is easy to read and can be a good addition to the literature exploring connections between fair learning and privacy. A couple of quick questions. I have read the Jagielski et al paper a while ago but the post-processing can be applied to any classifier, right? Furthermore, their setting can handle statistical parity, right? I would like the section regarding the connection to differential privacy to be a bit more detailed.

Reviewer 3



Section 1 describes the set-up of the problem. In particular, the authors emphasize that there are two cases where features might have noise in them: 1) when noise is deliberately added by researchers for privacy purposes and 2) in the "positive and unlabeled" setting where individual participants in the minority group might feel uncomfortable disclosing that, leading to unlabeled data for the sensitive feature in some cases. The case under consideration is binary classification on output $Y$ with a binary sensitive feature $A$. There are two main assumptions in this paper. The first is that the noise can be described as "mutually contaminated learning". In this case, each element in the distribution of corrupted $A=0$ examples is drawn from the true $A=0$ distribution with probability $\alpha$ and from the $A=1$ distribution with probability $1-\alpha$. Similarly, each element in the corrupted $A=1$ example is drawn from the true $A=1$ distribution with probability $\beta$ and from the $A=0$ distribution with probability $1-\beta$. The second assumption is that the fairness metric of interest falls under the category of "mean-difference scores", which means that they operate under the category of the mean difference in scores between the two subgroups. This paper uses two examples of such approaches (demographic parity and equality of opportunity) in its analysis. In Section 4 contains one of the main results of the paper. In particular, Theorem 2 shows that, for any given classifier $f$, the mean-difference score on the corrupted data is related to the mean-difference score on uncorrupted data by a simple scaling factor related to the degree of noise. This insight allows the authors to construct an algorithm to produce fair classification by incorporating existing tools. In particular, they rely on existing methods for estimating noise and finding fair classification algorithms in the absence of noise to produce fair classification algorithms in the presence of noise. Additionally, this section connects the idea of noise to that of privacy, though it's worth noting that this type of noise would only obscure the sensitive attribute $A$ and not the other features. Section 5 contains experimental results for classification in two scenarios, one where noise is added to the sensitive attribute for privacy purposes, and one where it is assumed that some members of the minority group fail to self-identify on an individual level. They compare their method to three baselines: 1) running the classifier on uncorrupted data 2) running it on corrupted data without accounting for this fact 3) a denoising method: the authors note that this undoes some of the privacy protections by inferring the sensitive label. In both cases, the authors show that their method achieves fairness and accuracy levels close to what would be obtained on uncorrupted data: a denoising method performs worse in terms of accuracy and has the added concern of violating privacy. While it returns lower fairness guarantees, it is debatable whether that is actually desirable, because it is returning lower guarantees than is desired, whereas the method this paper proposes is more consistent in what it returns. The paper is well-written and effectively communicates the ideas involved. Overall, this paper gives a fairly good treatment of what it set out to do. The main contribution is not very technically deep, but it does provide a simple and perhaps useful conceptual messsage: accounting for noisy labels can be achieved by tightening the tolerance proportional to the data quality. One minor point: the second footnote on page 3 states the assumption that the classifier $f$ doesn't use the sensitive attribute $A$. It might be useful to state in a more prominent portion of the text: without this caveat, the results seem counter-intuitive and make it harder for the reader to follow. EDIT: I have read the author response, and my vote to accept stands.

[Author Response · NeurIPS 2019]

Thanks for the valuable comments! Responses to each reviewer follow.

**R1:**

Our results in fact hold for most widely used group-based fairness definitions, including: (i) demographic parity and equality of opportunity (per Lines 120–124), (ii) equalized odds (by adding an additional constraint for $Y = 0$ in $\Lambda_D^{\mathrm{EO}}$), and (iii) disparate mistreatment, also known as accuracy parity (by using the 0-1 loss and $\Lambda_D^{\mathrm{DP}}$). Thus, we do not believe our results are narrow in scope.

> Non-binary sensitive features

Our theorem can be generalized to non-binary sensitive features, provided one makes stronger assumptions on the noise (e.g., that it is symmetric). Further developing this is certainly of interest, but studying a binary feature is a common starting point in both the fairness and label noise literature. As ours is the first study of the issue of noisy sensitive features (to our knowledge), we focused on getting a clean and practical result in this important case.

> MC noise model is restrictive

MC learning is an active, widely-used noise model (e.g., [1, 2, 3, 4, 5]). By ensuring that our technique works for this noise model, we have covered the two important and pervasive special cases of CCN [3, 1, 5] and PU learning [2, 4].

> Constraint $\alpha + \beta < 1$ in MC model is restrictive

The constraint imposes no loss of generality: when $\alpha + \beta > 1$, we can simply flip the two labels and apply our theorem. When $\alpha + \beta = 1$, all information about the sensitive attribute is lost. This pathological case is equivalent to not measuring the sensitive attribute at all.

> Unclear/undefined notation

The order of equations in (4) was accidentally swapped (our apologies). For all the other points raised: (i) $\Lambda_D$ is defined in the immediately following para (Lines 111-115) and examples are given in Equations 2 and 3; (ii) $\bar{L}$ is defined on Line 118, and $\bar{\ell}$ immediately before on Line 117; (iii) regarding redefinition of $L_D$, in both cases is the same quantity; the RHS in the second usage explicates the involvement of the sensitive attribute.

> (c) "In equation (5) ... given that $\alpha$ and $\beta$ don't sum up to 1, what happens to the remaining data points?"

There seems to be a slight misunderstanding: the mixture weights need only sum to 1 *within* and not *across* each corrupted class-conditional. One must take $\Pr(Y_{\mathrm{corr}} = 1)$ into account when reasoning about the samples. As a concrete example, consider a CCN setup where $\Pr(Y = 1) = \frac{1}{2}$, and +ve and –ves have a 0% and 50% chance respectively of having their label flipped. One may verify that in this case, $D_{1,\mathrm{corr}} = \frac{2}{3} \cdot D_1 + \frac{1}{3} \cdot D_0$ and $D_{0,\mathrm{corr}} = D_0$. Clearly, here $\alpha + \beta = \frac{1}{3} \neq 1$. The "missing" weight on $D_1$ is compensated by there being a greater fraction of corrupted +ves, as one can verify $\Pr(Y_{\mathrm{corr}} = 1) = \frac{3}{4}$. Indeed, the fraction of true +ves remains at $\frac{2}{3} \cdot \frac{3}{4} = \frac{1}{2}$, and so no sample goes missing.

**R2:**

> Post-processing of Jagielski et al. can be applied to any method, and can handle demographic parity

It is correct that the post-processing method of Jagielski et al. can be applied after any fairness-unaware learner. By contrast, our method can be applied to any *in-process* fairness-preserving learner (e.g. [Donini et al., 2018], [Agarwal et al., 2018], [Zafar et al., 2017b]) In-processing algorithms generally result in better tradeoffs than post-processing (e.g., [Agarwal et al., 2018]). We agree regarding demographic parity, and we will note this in our revision.

**R3:**

> Suggestions on expanding differential privacy discussion

We appreciate the insightful suggestions, and will expand our discussion accordingly. In particular, we do not foresee any difficulties in combining our approach with any privacy-preserving technique for standard (non-sensitive) features.

**References**

[1] N. Charoenphakdee, J. Lee, and M. Sugiyama. On symmetric losses for learning from corrupted labels. In *ICML*, 2019.
[2] S. Jain, M. White, and P. Radivojac. Recovering true classifier performance in positive-unlabeled learning. In *AAAI*, 2017.
[3] J. Katz-Samuels, G. Blanchard, and C. Scott. Decontamination of mutual contamination models. *JMLR*, 20(41):1–57, 2019.
[4] R. Kiryo, G. Niu, M. du Plessis, and M. Sugiyama. Positive-unlabeled learning with non-negative risk estimator. In *NIPS*. 2017.
[5] B. van Rooyen and R. C. Williamson. A theory of learning with corrupted labels. *JMLR*, 18(228):1–50, 2018.


[Meta-Review · NeurIPS 2019]

The author response adequately addressed concerns raised in the initial reviews.